# Therapeutic Potential of Albumin Nanoparticles Encapsulated Visnagin in MDA-MB-468 Triple-Negative Breast Cancer Cells

**DOI:** 10.3390/molecules28073228

**Published:** 2023-04-04

**Authors:** Abdullah Alsrhani, Abozer Y. Elderdery, Badr Alzahrani, Nasser A. N. Alzerwi, Maryam Musleh Althobiti, Musaed Rayzah, Bandar Idrees, Ahmed M. E. Elkhalifa, Suresh K. Subbiah, Pooi Ling Mok

**Affiliations:** 1Department of Clinical Laboratory Sciences, College of Applied Medical Sciences, Jouf University, Sakaka 72388, Saudi Arabia; 2Department of Surgery, College of Medicine, Majmaah University, Al-Majmaah 11952, Saudi Arabia; 3Department of Clinical Laboratory Science, College of Applied Medical Science, Shaqra University, Shaqra 11961, Saudi Arabia; 4Department of Surgery, Prince Sultan Military Medical City in Riyadh, Makkah Al Mukarramah Rd, As Sulimaniyah 12233, Saudi Arabia; 5Department of Public Health, College of Health Sciences, Saudi Electronic University, Riyadh 11673, Saudi Arabia; 6Department of Haematology, Faculty of Medical Laboratory Sciences, University of El Imam El Mahdi, Kosti 1158, Sudan; 7Centre for Materials Engineering and Regenerative Medicine, Bharath Institute of Higher Education and Research, Chennai 600073, India; 8Department of Biomedical Science, Faculty of Medicine & Health Sciences, Universiti Putra Malaysia, Serdang UPM 43400, Malaysia

**Keywords:** breast cancer, albumin visnagin nanoparticles, MDA-MB-468 cell line, apoptosis, nanotechnology

## Abstract

Breast cancer is among the most recurrent malignancies, and its prevalence is rising. With only a few treatment options available, there is an immediate need to search for better alternatives. In this regard, nanotechnology has been applied to develop potential chemotherapeutic techniques, particularly for cancer therapy. Specifically, albumin-based nanoparticles are a developing platform for the administration of diverse chemotherapy drugs owing to their biocompatibility and non-toxicity. Visnagin, a naturally derived furanochromone, treats cancers, epilepsy, angina, coughs, and inflammatory illnesses. In the current study, the synthesis and characterization of albumin visnagin (AV) nanoparticles (NPs) using a variety of techniques such as transmission electron microscopy, UV-visible, Fourier transform infrared, energy dispersive X-ray composition analysis, field emission scanning electron microscopy, photoluminescence, X-Ray diffraction, and dynamic light scattering analyses have been carried out. The MTT test, dual AO/EB, DCFH-DA, Annexin-V-FITC/PI, Propidium iodide staining techniques as well as analysis of apoptotic proteins, antioxidant enzymes, and PI3K/Akt/mTOR signaling analysis was performed to examine the NPs’ efficacy to suppress MDA-MB-468 cell lines. The NPs decreased cell viability increased the amount of ROS in the cells, disrupted membrane integrity, decreased the level of antioxidant enzymes, induced cell cycle arrest, and activated the PI3K/Akt/mTOR signaling cascade, ultimately leading to cell death. Thus, AV NPs possesses huge potential to be employed as a strong anticancer therapy alternative.

## 1. Introduction

Women are more likely than men to be diagnosed with breast cancer [1]. In women, it is the most prevalent tumor and is classified as the second major cause of cancer death. An inner lining of the milk ducts and breast tissues is affected by this lobule malignancy. The two subtypes are human epidermal growth factor receptor-2 negative and hormone receptor-positive breast carcinomas. In developed nations, breast cancer is detected at 80%, and in developing countries at 40% [2,3].

Nanotechnology is a promising discipline that has generated numerous opportunities in fields including materials science and health care [4]. Nanoparticles outperform their bulk counterparts in terms of their nanometer size, substantial surface area, increased surface reactivity, enhanced physicochemical features, and large surface-to-volume ratio. These characteristics endow them with unique physical qualities, opening up new avenues for early detection, effective therapy, and the diagnosis of numerous human diseases, including cancer [5]. Nanoparticle-based therapeutic technologies, specifically drug delivery systems, have emerged as prominent study subjects [6].

Human serum albumin, the most prevalent physiological protein present in plasma, is a suitable nanocarrier for drug administration owing to its excellent biocompatibility, cheap cost, extended plasma half-life, and mass production [7]. Since albumin readily increases the apparent solubility of hydrophobic smaller compounds in plasma and influences their effective transport to the target site, this protein has the capacity to play a significant function in drug transport as well as potency [8]. NPs based on albumin have several benefits in this respect, notably biocompatibility, non-toxicity, and easy surface modification because of the presence of amino and carboxylic groups [9,10]. The increased numerous binding sites allow for the incorporation of hydrophilic and hydrophobic molecules in the nanoparticle framework. Thus, the emergence of albumin-based nanocarriers for drug delivery is promising [11].

Secondary metabolites derived from plants are less harmful and include lead components helpful in drug development. *Ammi visnaga* L. is among the most beneficial plants in this regard due to its chemical composition, which consists primarily of furanochromone compounds [12]. Furanochromones are well-established oxygen-containing heterocyclic molecules that play biologically significant roles in nature. For their vasodilating and antispasmodic properties, *A. visnaga* fruits, khellin, and visnagin have been extensively studied [13]. Moreover, visnagin has been utilized to treat several tumors, epilepsy, angina, cough, inflammatory diseases, gall bladder problems, and renal discomfort. It is regarded as a robust cardiac vasodilator, as well as having a role in the treatment of bronchial asthma [14,15,16].

In this study, a first-time attempt at combined synthesis of albumin visnagin nanoparticles to assess their synergism for the inhibition of breast cancer cells has been performed. Following the synthesis of the NPs, their characterization through various techniques is conducted to confirm their formation. Further, their anticancer activities have been determined.

## 2. Results and Discussion

### 2.1. Spectral Characterization of Albumin-Binding Nanoparticles

In the current investigation, the synthesis, and characterization of albumin-binding nanoparticles were performed using a variety of techniques. Figure 1a shows the results of albumin-binding nanoparticle spectroscopic analysis with UV-Vis spectroscopy. Using a UV-Vis spectrometer, the optical characteristics of the fabricated albumin-visnagin NPs were investigated. UV-visible wavelengths are 200 to 1100 nm. The UV-visible absorption spectra of proteins bound to nanomaterials undergo certain modifications, including a modest spectral shift, a minor spectral broadening, and minute variations in the intensity of the spectra [17]. The UV-Vis spectroscopy absorbance band at 317 nm confirmed albumin-derived visnagin NPs.

As seen in Figure 1b, FTIR spectroscopy is an appropriate method for assessing albumin visnagin NP surface functionality. The albumin visnagin NP’s FTIR spectra revealed several unique bands at 3427, 2924, 2846, 1646, 1549, 1389, 1095, 847, and 610 cm^−1^, respectively. The stretching vibration of the O-H groups is responsible for the broadband detected at 3427 cm^−1^ [18]. The bands between 1700 and 1500 cm^−1^ received extra attention. In the case of proteins, this is where secondary structure-related absorption bands are expected to appear. It was revealed that the band at around 1646 cm^−1^ might be accredited to the stretching of the C-O mode present in the amide-I band. Furthermore, a vibration-induced absorption band for C-N stretching is found at 1549 cm^−1^. The functional groups creating the peptide link between albumin amino acids have both amide I and amide II bonds [19]. The peaks at 2924 and 2846 cm^−1^ contribute to asymmetric and symmetric C-H stretching. The peak observed at 1389 cm^−1^ may have C-C or C-N stretching vibrations [18]. The peak at 1095 cm^−1^ is due to the stretching vibrations of C-O-C present in the visnagin. The C-H bending for the benzene ring is shown by the absorbance band obtained between 837 and 721 cm^−1^ [20]. The PL emission spectra of the albumin visnagin NPs are shown in Figure 1c. Albumin visnagin NP’s exhibited PL emission at 431, 449, 481, and 513 nm. Violet emission occurs at 431 nm, blue emission occurs at 449 and 481 nm, and followed by green emission at 513 nm.

The broad peaks obtained in the X-Ray diffraction (XRD) spectrum of albumin visnagin NPs confirmed that the NPs were amorphous in nature (Figure 1d). This reduction in crystallinity indicates that the amorphous characteristics of visnagin were increased within the albumin nanoparticles [21]. DLS was utilized to determine the hydrodynamic diameter of the albumin visnagin NPs, as shown in Figure 1e, which is determined to be 149.30 nm. The hydrodynamic diameter, also known as the hydrodynamic size, is different from the physical size since the NPs were surrounded by a water medium.

The size, morphology, and crystalline nature of the nanostructured particles were evaluated using TEM imaging methods and SAED (selected area electron diffraction) pattern analysis. Figure 2a–d shows a TEM image of artificial albumin visnagin NPs. Low-resolution TEM was used to observe extremely tiny nanoparticles. The higher magnification images clearly show the spherical structure produced by the NPs (Figure 2c). SAED was carried out for the albumin and visnagin NPs presented in Figure 2d. The hybrid nanomaterial sample made from albumin and visnagin displays an SAED pattern of tiny single crystals and polycrystalline rings.

Figure 2e,f display the FESEM images of the albumin visnagin NPs at lower and higher magnifications. As shown in Figure 2e,f, while operating as albumin visnagin NPs, the visnagin nanoparticles are dispersed unevenly or randomly, interacting with albumin. FESEM images clearly show the spherical structure of albumin visnagin NPs and the presence of aggregation. The size of the NPs was 120 nm. The EDAX spectra of albumin visnagin NPs are presented in Figure 2g. It was evident that albumin visnagin NPs were generated during the EDAX study due to C, N, S, and O-related peaks.

### 2.2. Cytotoxic and Apoptotic Effect of Albumin-Binding Nanoparticles in MDA-MB-468 Cells

The influence of generated NPs on MDA-MB-468 cell viability is shown in Figure 3 by the MTT assay. Following incubation for 24, 48, and 72 h with NPs at doses of 2.5, 5, 10, 20, 40, 80, and 160 µg/mL, it was observed that NP treatment reduced cell viability dose-dependently. For 72, 48, and 24 h, the IC50 was 30.56, 24.28, and 17.8 g/mL, respectively. The concentrations of IC50 (17.8 µg/mL) and IC25 (8.9 µg/mL) obtained from 24 h were selected for further investigation. The findings demonstrated that albumin and visnagin NPs dose-dependently inhibit the cell viability of MDA-MB-468 cells. This is due to the visnagin’s action. Beltagy and Beltagy, 2015 [22] reported that visnagin exhibited more potency against HCT-116 and Hela cell lines. It also attacked MCF-7 cell lines.

Figure 4 depicts the effect of albumin visnagin on MDA-MB-648 cell apoptosis by AO/EB dual staining. AO is a stain that colors the nuclei of both viable and dead cells green, while EB only colors the nuclei of cells with reduced membrane integrity red (EB). As a result, early apoptotic cells appear reddish red while viable cells appear green. Late apoptotic cells are indicated by red hues. According to the findings, the lower and higher concentrations of the nanoparticle showed yellowish-red and red staining together with a decline in the green staining of the nuclei, which suggested cell damage and apoptosis. In many tissues, the natural process of cell death known as apoptosis may be seen. It is distinguished by particular morphological traits and extensive DNA fragmentation [23]. To assess the cell death approach produced by NPs, the morphological alterations associated with apoptosis were examined using the AO/EB staining method. Acridine orange diffusion into living cells’ cellular membranes caused them to fluoresce green. However, treated cells underwent apoptosis and fluoresced orange due to nuclear shrinkage. Oxidative stress is a crucial factor that results in cell injury and plays a major role in cancer etiology. The antioxidant system removes these free radicals, helping cells keep oxidative stress in check. Visnagin’s antioxidant property aids in apoptosis [24].

The impact of albumin visnagin NPs on cell cycle analysis using PI staining is seen in Figure 5. The DNA composition of untreated, nanocomposite-treated, and doxorubicin-treated MDA-MB-468 cells was assessed using flow cytometry. In comparison to untreated cells, nanocomposite-treated cells’ DNA content analysis exhibited a sub-G1 phase. Our findings demonstrated a substantial decrease in the viability of significant cell suppression against MDA-MB-468 cell lines after a 48 h treatment period with the selected IC50 dose. Flow cytometry identified cell cycle arrest phases, and it was discovered that the exposed cells demonstrated more sub-G1 cells than the control cells. Doxorubicin and the generated NPs markedly reduced the proportion of cells arrested at the G2/M phase compared to untreated cells. Hence, cell cycle arrest occurs at the G2/M stage, which is analogous to doxorubicin’s impact on cells. Cell cycles were investigated using propidium iodide (PI) staining. As PI is less costly, more stable, and better at predicting cell viability since it rejects dyes in live cells, it is used more frequently than other staining. Membrane permeability determines whether PI may enter a cell and because the plasma membrane is undamaged, PI cannot identify live or early apoptotic cells [25]. Many stressors are associated with intracellular ROS generation, which may facilitate cell cycle arrest or cellular death. DNA damage is a biological event closely associated with cell cycle arrest [26]. In comparison to control cells, doxorubicin, and albumin visnagin NPs at the IC50 concentration generate a greater number of G2/M-stage cells in the treated cells. Therefore, cell cycle arrest occurred at the G2/M phase. In MDA-MB-468 cells, the nanoparticle dramatically halted the cell cycle. A similar study found that gold nanoparticles (GNPs) and their conjugates (GNPs-PEG-l-asparaginase-RGD) induced apoptosis and inhibited the cell cycle at the G2/M stage. By upregulating pro-apoptotic p53 while decreasing anti-apoptotic Bcl-2, MCF-7 cells secreted cytochrome C more due to the mitigation of mitochondrial membrane potential [27].

Figure 6 shows the impact of albumin visnagin NPs on MDA-MB-468 cells’ apoptosis. Cells in the upper quadrant of the cells were necrotic and late apoptotic by Annexin V-FITC/PI staining. In the lower left quadrant, Annexin V-FITC and PI were not used to stain normal living cells. In the lower right quadrant of the cells, Annexin V-FITC labeling only revealed early apoptotic cells. NP-exposed cells had more early apoptosis than control. Similar patterns were present in the population that had undergone late apoptosis. This is suggestive of the considerable apoptotic potential of albumin visnagin NPs. An essential step in the apoptotic process is the translocation of phosphatidylserine (PS) from the inner membrane layers to the outer surface. A fluorescent probe called Annexin V-FITC binds to PS Ca^2+^ independently. The PS molecules can bind to the Annexin V-FITC conjugate protein after they reach the membrane surface. Moreover, PI vital staining is used to evaluate membrane integrity to check if it has significantly deteriorated or been preserved [28]. Annexin V/PI staining was employed to ascertain the apoptotic level brought on by albumin visnagin NPs. From a time- and dose-based perspective, the flow cytometric study of treated cells in the tested cells revealed a drastic decline in the fraction of live cells. Just a tiny percentage of cells died of necrosis. This might be due to the removal of necrotic cells, formed by late-stage apoptotic cells that have lost the ability to repair DNA [29].

Albumin visnagin NP treatment increased DNA strand breaks significantly after 12 h. The results (Figure 7c) show that the maximum damage induced by the comet assay was reached after 24 h and that it was dose-dependent. In undamaged cells, nuclei lacked tails, while damaged cells had comet-like nuclei (Figure 7b). DNA damage detected in albumin visnagin NPs was substantially higher than in the control (Figure 7a). Comets of classes 3 and 4 were observed to be more prevalent in the albumin visnagin NPs treated group than in the control. The findings were noted with undamaged cells; comet class 0 was detected more frequently in the control than in the albumin visnagin NPs treated group (Figure 7d).

The effect of albumin visnagin NPs on the level of ROS in MDA-MB-46 cells as determined by the DCFH-DA staining technique is shown in Figure 8. Control cells (Figure 8a) showed slight fluorescence; however, when the IC25 or IC50 of NPs was introduced before DCFH-DA, a significant fluorescence increase was observed. It demonstrated that NPs treatment caused an upsurge in endogenous ROS by revealing an increase in fluorescence spike when compared to control cells. Our data suggest that albumin visnagin NPs particularly target cancer cells, most likely because these cells have significant ROS concentrations.

Aging and death of tumor cells can result from an unbalanced, abrupt rise in intracellular ROS [30]. One of the most popular methods for examining intracellular ROS formation involves the use of fluorescent dyes, such as DCFH-DA. As a result, the intensity of the fluorescence is directly relational to the quantity of peroxide generated by the cells [31]. Using MDA-MB-468 cell lines, the current study assessed the capacity of albumin visnagin NPs to produce ROS. The IC50 concentration of NPs was observed to increase ROS generation in the exposed cells, as seen by increased green fluorescence.

### 2.3. Effect of Albumin Visnagin NPs on Antioxidant Enzymes in MDA-MB-468 Cells

In control and IC25 and IC50 treated MDA-MB-468 cells, the status of antioxidants such as SOD, CAT, GSH, and MDA were measured. The findings are shown in Figure 9. Following NP treatment, it was discovered that the levels of malondialdehyde (MDA) in treated cells were much greater. In addition, the levels of SOD, CAT, and GSH dropped. This indicated that the fabricated NPs had a carcinogenic impact on breast cancer cells. By combining ROS with antioxidants, oxidative stress accelerates the development of diseases such as breast cancer. The antioxidant enzymes CAT, GPX, and SOD are all part of the body’s sophisticated antioxidant defense system. They prevent the start of chain reactions involving free radicals. Free radicals may trigger an array of reactions by interacting with macromolecules. This can lead to cellular malfunction and even death when they are produced in excess or when the cellular antioxidant mechanism is weak. The SOD and CAT enzymes play a pivotal function in the management of the disease of breast cancer [32]. The amounts of SOD, CAT, and GSH in cells treated with NPs diminished, but the levels of MDA enzyme were elevated, indicating that the NPs caused these cells to undergo oxidative stress-mediated cell death.

### 2.4. Effect of Albumin Visnagin NPs on Expression of Pro and Anti-Apoptotic Proteins in MDA-MB-468 Cells

The parameters were estimated using the ELISA method to determine if albumin visnagin NPs-induced apoptosis on MDA-MB-468 cells involves the activation/repression of apoptotic markers. Figure 10 displays the results of the examination into how the nanoparticle treatment affected the status of Bax, Bcl-2, Cyt-C, and p53 as well as caspase-3, -8, and -9. After receiving IC25 and IC50 concentrations of NPs, the activities of caspase-3, -8, and -9, as well as Bax, p53, and Cyt-C, were markedly increased in the breast cancer cells. Nevertheless, the nanoparticle treatment caused Bcl-2 levels to drop, demonstrating its anti-carcinogenic potential (Figure 10). Caspases are a group of cysteine proteases that includes effector caspases such as caspase-3 and -7 as well as initiator caspases such as caspase-8 and caspase-9. The intrinsic apoptosis signaling cascade’s last executioner of cell death, caspases are cleaved in response to drug-induced mitochondrial stress [33]. Extracellular cell death-inducing ligands connect to death receptors on the plasma membrane to create the cell death signaling complex. As a result, initiator caspases are activated, which cleave and activate effector caspases later. The intrinsic signaling is activated when the mitochondrial membrane ruptures, releasing apoptogenic agents such as cytochrome c [34]. The parameters were estimated using the ELISA technique in order to determine whether the apoptosis that albumin visnagin NPs elicited in MDA-MB-468 cells was caused by activation or inhibition of apoptotic markers. After receiving IC25 and IC50 concentrations of NCs, the liver cancer cells significantly increased their caspase-3, -8, and -9 activities, as well as Bax, p53, and Cyt-C. Nevertheless, the application of the nanocomposite confirmed its anti-carcinogenic potential by causing a drop in Bcl-2 levels.

The expression patterns of the proteins involved in cell signaling (PI3K, AKT, and mTOR) were investigated using RT-PCR (Figure 11). We looked at two different levels of albumin visnagin NPs (IC25 and IC50). High concentrations of NPs prevent cell growth, which results in the cell proceeding through apoptosis. The IC50 concentration was found to increase the number of apoptotic proteins and decrease the PI3K/AKT/mTOR expression in breast cancer cells. As a result, NP therapy may have started the apoptotic pathway in breast cancer cells. Pathological and physiological processes require the proper PI3K/Akt/mTOR signaling [35]. The majority of human cancers have been shown to overexpress mTOR and activate the PI3K/Akt/mTOR cascade [36]. As a consequence, in the present investigation, NPs efficiently blocked the PI3K, AKT, and mTOR expressions. This demonstrates that NPs effectively decreased the growth of breast tumor cells and induced death.

The molecular formation and anticancer properties of albumin visnagin NPs have been confirmed through the chemical synthesis and chemical characterization of these nanoparticles using MDA-MB-468 cells. In MDA-MB-468 cell lines, albumin visnagin NPs exhibit a sphere-shaped structure and a larger average crystallite size, which increases their cytotoxicity. According to MTT testing, the dual AO/EB assay, DCFH-DA, Annexin-V/FITC/PI, and PI staining, the nanomaterials showed significant cytotoxicity toward MDA-MB 468 cells. By using a comet assay, albumin visnagin NPs resulted in increased ROS levels and nuclear damage in the cells. The PI3K/Akt/mTOR signaling cascade was activated, membranes were disrupted, and antioxidant enzyme levels were reduced, resulting in the death of cells. A strong anticancer therapy alternative, albumin visnagin NPs, should be investigated as part of future research.

## 3. Materials and Methods

### 3.1. Chemicals

Sigma Aldrich provided visnagin, human serum albumin, glutaraldehyde, and other chemicals (St. Louis, MO, USA). All of the chemicals purchased are analytical grade from Sigma Aldrich (St. Louis, MO, USA).

### 3.2. Synthesis of Albumin Visnagin NPs

For 6 h at room temperature, 50 mg visnagin was mixed with 500 mg human serum albumin in 20 mL of distilled water. After that, it was cross-linked with 0.5% glutaraldehyde (100 mL). The unreactive components were then removed by dialysis in water for one day, resulting in the formation of albumin visnagin NPs.

### 3.3. Characterization Analysis of Albumin Visnagin NPs

The obtained albumin visnagin NPs sample was characterized using an X-ray diffractometer (XRD) (X’PERT PRO PANalytical). With a monochromatic CuK diffraction beam of wavelength 1.5406, the diffraction patterns for albumin visnagin NPs were recorded in the 2ϴ range of between 25° and 80°. The albumin visnagin NPs sample was examined using a FESEM (Carl Zeiss Ultra-55 FESEM) with EDX Spectrometry (model: Inca). The morphologies of the NPs were examined with a TEM (Tecnai F20 model) instrument set to an accelerating voltage of 200 kV. The Perkin–Elmer spectrometer was utilized to record the FTIR spectrum in the wavenumber range 400–4000 cm^−1^. The Lambda-35 spectrometer was utilized to measure the absorption spectra of NPs between 200 and 1100 nm. Photoluminescence (PL) spectra were obtained using a Perkin–Elmer-LS 14 spectrophotometer [37,38,39].

### 3.4. Cell Culture Chemicals

Human breast cancer MDA-MB-468 cells were purchased from Sigma Aldrich (St. Louis, MO, USA). The cells were grown using DMEM from Sigma-Aldrich (St. Louis, MO, USA). It contained 2% L-glutamine, D-glucose, hypoxanthine monosodium salt, lipoic acid, linoleic acid, putrescine dihydrochloride, pyruvate, thymidine, phenol red indicator, vitamins, sodium amino acids, and 2% sodium bicarbonate. It also contained 100 µg/mL streptomycin, 5% FBS, and 100 µg/mL penicillin from Sigma-Aldrich (St. Louis, MO, USA).

### 3.5. Cell Lines

The NCCS, Pune provided us with the MDA-MB-468 and MDA-MB-231 human breast cancer cells. A 10% FBS-supplemented DMEM (Thermo Fisher Scientific Inc., Waltham, MA, USA) medium was used for all cell lines in addition to penicillin, streptomycin, and glutamine.

### 3.6. Cell Viability Assay

Utilizing the MTT test, the cytotoxicity of albumin visnagin NPs was evaluated. The cells were loaded into a 96-well plate at 1 × 10^5^ cell population per mL in DMEM medium without serum for a duration of 24–48 h at 37 °C and 5% CO_2_. The wells were incubated with different quantities of NPs (2.5, 5, 10, 20, 40, 80, and 160 µg/mL) in a DMEM medium for 24, 48, and 72 h after being rinsed with sterile PBS. The analysis was carried out three times. Following the first incubation, cells were exposed to MTT (5 mg/mL) for an additional 2–4 h until purple clusters were readily evident using an inverted microscope. MTT was drained out of the wells and cleaned with 200 µL of 1X PBS. A total of 100 µL of DMSO was added to the plate, which was then agitated for 5 min to dissolve the formazan crystals. GraphPad Prism 6.0 software was utilized to determine the percentage of cell viability and IC25 and IC50 value for each well after the absorbance was estimated at 570 nm [40].

### 3.7. Determination of Cell Apoptosis by AO/EtBr Dual Staining

With the use of dual staining, the MDA-MB-468 cells were marked and easily separated from the live and necrotized cells. The IC50 concentration of albumin visnagin NPs along with Paclitaxel at 0.5 µM concentration was applied to the cells, which were grown in a 6-well plate at 3 × 10^4^ cell population per well. For 24 h, the cells were maintained. After being pretreated with glacial acetic acid: methanol (1:3) at 4 °C for 30 min, cells were subjected to a 1:1 combination of AO/EtBr for 30 min at RT. Fluorescence microscopy was used to capture the cells at a magnification of 20X [40].

### 3.8. Cell Cycle Analysis by PI Staining

PI staining was utilized to examine the cell cycle. Cells exposed to IC50 concentration of albumin visnagin NPs for 48 h and standard drug Doxorubicin (5 µM/mL) were trypsinized, then washed in PBS and fixed in 90% ethanol. Following two PBS rinses, fixed cells were stained for 1 h in 50 μM PI with 5 μg/mL DNase-free RNase. The results were then analyzed using Cell Quest software and flow cytometry [41].

### 3.9. Assessment of DNA Damage by Comet Assay

DNA damage was examined using the comet assay. Initially, 10,000 cells from each sample of nanoparticle-treated (low and high concentration), and control cells were divided into aliquots, placed in tubes, and centrifuged at 800× *g* for 10 min at 4 °C. The cells were re-suspended in 0.7% agarose after the supernatant was removed, and 0.035 mL of this solution was applied to pre-coated and high-throughput comet assay slides. These slides have a silicon barrier between them so that 10 distinct samples can be layered on each slide at once. In order to improve adhesion, the clean portions are produced with a dried agarose covering. At 4 °C, the microgels on the slides were left to set. To lyse the attached cells and enable DNA to unfold, the slides were then submerged in ice-cold, lysis solution overnight in the dark at 4 °C. The slides were left in an alkaline buffer for 30 min after being incubated in the lysis solution to allow the DNA to unwind. Following that, electrophoresis was performed in the same buffer for 20 min at 1 V/cm. Each slide’s DNA was stained with 15 µL of EtBr (20 µg/mL) and then monitored under a fluorescence microscope after neutralizing in a Tris buffer (pH 7.5) and dehydrating in 75% methanol [42].

For 5 min, 20 mL of Ethidium bromide dye (20 µg/mL) was added to the slides. Using the Nikon Eclipse 80i fluorescent microscope equipped with a 450–490 nm excitation filter, the images were viewed at 200x using coverslips. The DNA damage of 100 cells per sample was calculated visually by analyzing comet appearance in 100 random cells. There are five classes of comets: one with no damage or less damage (5%); two with mild damage (5–20%); three with mid damage (20–40%); four with more damage (40–75%); and five with extreme damage (75%) [25].

### 3.10. Determination of Cell Apoptosis by Annexin V-FITC/PI Staining

Using Annexin V staining, the degree of apoptosis was assessed. A 24-well plate was used to culture MDA-MB-468 cells, which were then incubated for a duration of 24 h at 37 °C in CO_2_ (5%). Following the first incubation, each well received a medium containing the IC50 of albumin visnagin NPs, which was then incubated at a temperature maintained at 37 °C and 5% CO_2_. After that, cells were harvested, rinsed with PBS, and labeled with 5 µL of annexin V-FITC/PI (1 mg/mL). To assess the tagged cells, flow cytometry was employed [43].

### 3.11. Analysis of Reactive Oxygen Species (ROS) by DCFH-DA Staining

To examine the ROS formation, 2,7-dichlorofluorescein diacetate (DCFH-DA) was employed in line with the supplier‘s guidelines. Briefly, MDA-MB-468 cells were cultured (5 × 10^3^ cells/well) and then treated with or without N-acetylcysteine for 1 h at 37 °C, followed by exposure to the NPs at concentrations of IC25 and IC50 after 48 h at room temperature. The untreated and treated cells were harvested, centrifuged at 600× *g* for 4 min at 37 °C, rinsed in PBS, re-suspended in PBS with 10 µM DCFH-DA, and incubated for a duration of 15 min in the dark condition at 37 °C. Then, cells were cleansed with PBS before being immediately assessed with flow cytometry to check for the development of the fluorescent oxidized derivative of DCFH-DA at an emission and excitation wavelength of 525 nm and 450 nm, respectively [44].

### 3.12. Analysis of Oxidative Stress Parameters

After the cells had been treated with NPs in accordance with the instructor’s procedure, the SOD assay commercial kit (Superoxide Dismutase Activity Assay kit, Colorimetric, cat no = ab65354, from Abcam, Cambridge, UK) was used to measure SOD activity. The samples’ absorbance was determined at 450 nm with the help of a plate reader. Following the kit instructions, the CAT assay kit was utilized to investigate the activity of the CAT enzyme. Based on the breakdown of hydrogen peroxide, the enzyme is measured with the kit (Catalase Activity Assay Kit, Colorimetric/Fluorometric cat no = ab83464, from Abcam, Cambridge, UK). The levels of MDA activity were also measured using a kit. At 450 nm, the samples’ absorbance was measured. MDA and thiobarbituric acid combine to produce a complex, after which the absorbance was measured at 532 nm to assess lipid peroxidation (Lipid Peroxidation (MDA) Assay Kit (Colorimetric) cat no = ab233471, from Abcam, Cambridge, UK). The levels of GSH activity were also measured using a kit. Using a kinetic assay, the amount of GSH is measured by continuously reducing 5,5′-dithiobis (2-nitrobenzoic acid) in the presence of catalytic quantities (nmoles) of GSH (GSH Assay Kit, Colorimetric, cat no = ab239709, from Abcam, Cambridge, UK).

### 3.13. Measurement of Caspase-3, -8, -9, Cyt-C, p53, Bax, and Bcl-2 by ELISA

To further confirm the apoptotic cell death potential of the albumin visnagin NPs, different effectors of apoptosis were measured. Commercial ELISA assay kits (Abcam, Cambridge, UK) were utilized to estimate Caspase-3, -8, -9, Cyt-C, p53, Bax, and Bcl-2 protein status in albumin visnagin NPs-treated MDA-MB-468 cells at two different concentrations (IC25 and IC50).

### 3.14. Quantitative Real-Time PCR Analysis

The mRNA expression of the PI3K/Akt/mTOR signaling proteins was examined in MDA-MB-468 cancer cells that received NPs treatment. Utilizing a commercially available RNAEasy kit, the RNA was isolated from the control cells and the treated cells (Qiagen, Germany). The manufacturer’s protocol was followed for the isolation procedures. The Nanodrop spectrophotometer was employed to measure and evaluate the quantity and quality of RNA. The Sybr green PCR kit was utilized for the PCR process to analyze the PI3K/Akt/mTOR proteins. Data on the relative intensity of the targeted genes were collected and examined. With reference to earlier publications, the primer sequences for the targeted genes were retrieved from the PUBMED sequence (Table 1). The GAPDH was utilized as an internal housekeeping gene to standardize the mRNA expressions of the target markers, and the relative expressions were calculated using the 2^−ΔΔCT^ formula [45].

### 3.15. Statistical Analysis

The mean and standard deviation of three separate triplicate trials is represented. The significance of variances between the treated and control groups was assessed using one-way ANOVA, and the outcome was then subjected to a student’s *t*-test. To signify statistical significance, a *p*-value of less than 0.05 was employed.

## 4. Conclusions

In conclusion, albumin visnagin NPs have been synthesized and characterized through various techniques to confirm their formation. The average crystallite size of the NPs was reported to be 120 nm, and it was discovered that the NPs have a spherical structure. Furthermore, the MTT test, dual AO/EB, DCFH-DA, Annexin V-FITC/PI, and Propidium iodide staining techniques demonstrated the nanomaterials’ considerable cytotoxicity against MDA-MB-468 cells. The research showed that the NPs increased the amounts of ROS in the cells, disrupted the membrane integrity, decreased the level of antioxidant enzymes, induced cell cycle arrest, and activated the PI3K/Akt/mTOR signaling cascade, ultimately leading to cell death. Thus, albumin visnagin NPs offer great potential to be employed as a strong anticancer therapy alternative. Future research will focus on analyzing the molecular mechanism of in vivo cell death in breast cancer cells.

## Figures and Tables

**Figure 1 molecules-28-03228-f001:**
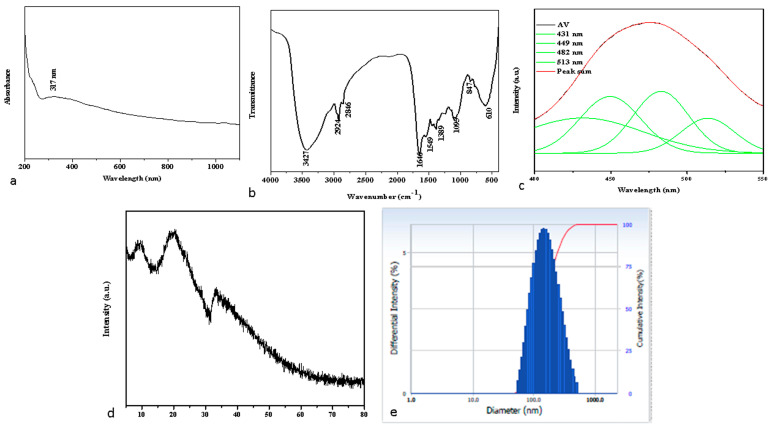
Characterization of albumin visnagin NPs. UV-Vis spectrophotometer (**a**), FTIR Transmittance vs. wavenumber chart (**b**), and PL spectrum (**c**) analysis of synthesized albumin visnagin NPs. (**d**) X-ray diffraction Pattern and (**e**) DLS spectrum of albumin visnagin NPs.

**Figure 2 molecules-28-03228-f002:**
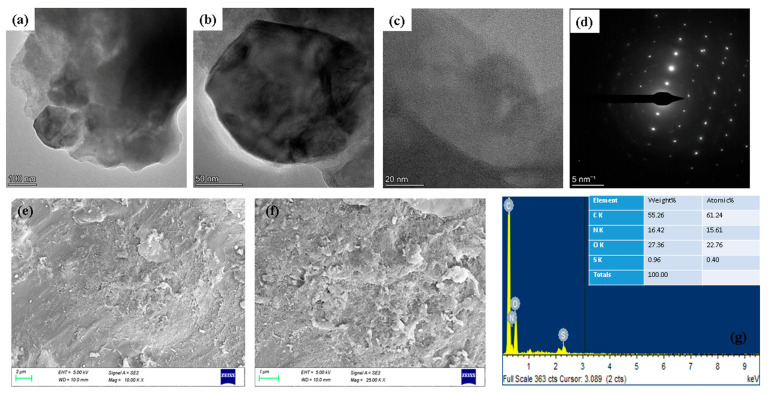
Electron microscopic imaging of albumin visnagin NPs lower and higher magnification TEM images (**a**–**c**) and SEAD patterns (**d**). Lower and higher magnification FESEM (**e**,**f**) and EDAX (**g**) spectrums.

**Figure 3 molecules-28-03228-f003:**
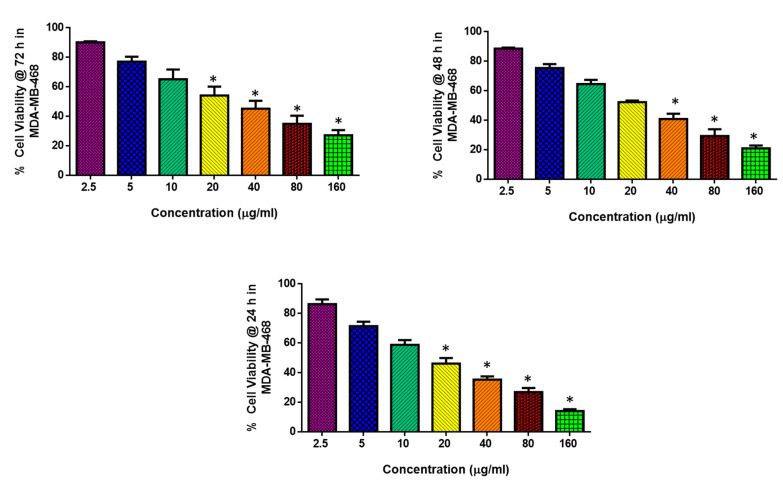
Albumin visnagin NPs cause cytotoxicity in MDA-MB-468 cells. Cells were treated with various concentrations (2.5–160 µg/mL) of albumin visnagin NPs for 24, 48, and 72 h and the viability was assessed by MTT assay. Data are illustrated as mean ± SD of triplicates. * Signifies *p* < 0.05 when compared with control.

**Figure 4 molecules-28-03228-f004:**
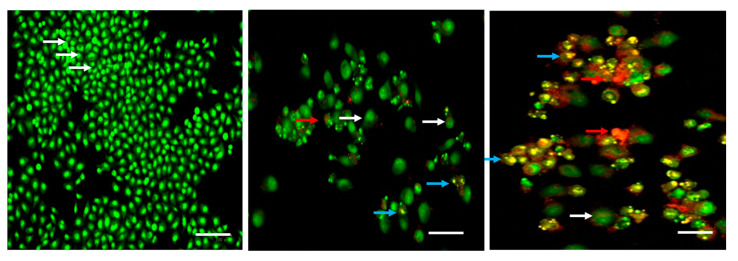
Effect of albumin visnagin NPs on apoptosis in MDA-MB-468 cells for 24 h. The cells were stained with ethidium bromide and acridine orange (1:1) and assessed using fluorescence microscopy (Labomed, Los Angeles, CA, USA). As shown by the white arrow, the control cells exhibited green fluorescence (indicating viable cells). As shown by the blue arrow, albumin visnagin NPs-treated cells revealed yellow/orange fluorescence, indicating early and late apoptosis, respectively. Necrosis is indicated by the red arrow. Control (untreated cells) (**Left**), albumin and visnagin NPs-treated cells; IC25 (**Middle**) and IC50 concentrations (**Right**). This is a representative image of the experiment performed in triplicate with 20X magnification (scale bar = 100 mm).

**Figure 5 molecules-28-03228-f005:**
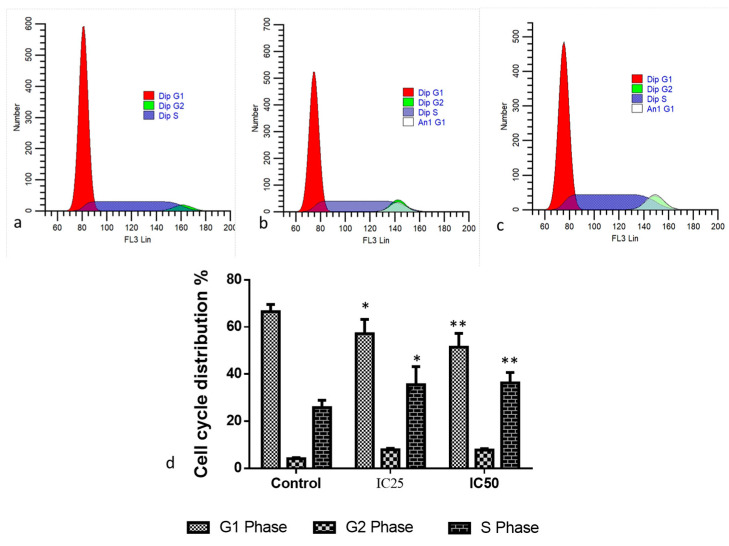
Cell cycle analysis was performed using flow cytometry after staining with Propidium Iodide (PI). Albumin and visnagin NPs were administered to MDA-MB-468 cells at IC50 concentrations for 48 h compared with controls. An analysis of cell cycle patterns and apoptosis distribution in untreated or control (**a**) and albumin and visnagin NP-treated cells, and their concentrations of IC25 (**b**) and IC50 (**c**). Percentage of cell cycle distribution (**d**). (* signifies *p* < 0.05 when compared with control, ** signifies *p* < 0.001 when compared with control).

**Figure 6 molecules-28-03228-f006:**
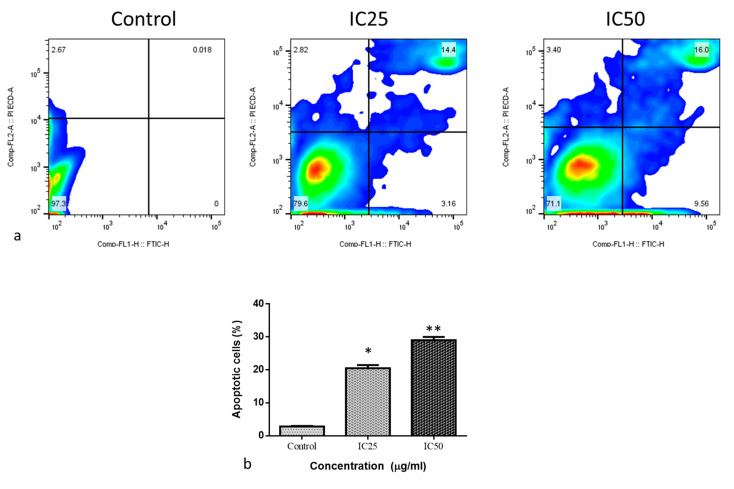
Flow cytometry analysis of MDA-MB-468 cancer cells. The cells were exposed to IC50 concentration of albumin visnagin NPs for 48 h. At least two independent experiments were conducted to produce these figures. Live cells, early apoptotic, and necrotic cells were revealed in the lower left quadrant, the lower right quadrant, and the upper (Annexin-V+/PI+) quadrant, respectively (**a**). After albumin visnagin NP administration to MDA-MB-468 cells, the proportion of early and late apoptotic cells was determined (**b**). Two independent experiments were conducted to obtain the mean ± SD. (* signifies *p* < 0.05 when compared with control, ** signifies *p* < 0.001 when compared with control).

**Figure 7 molecules-28-03228-f007:**
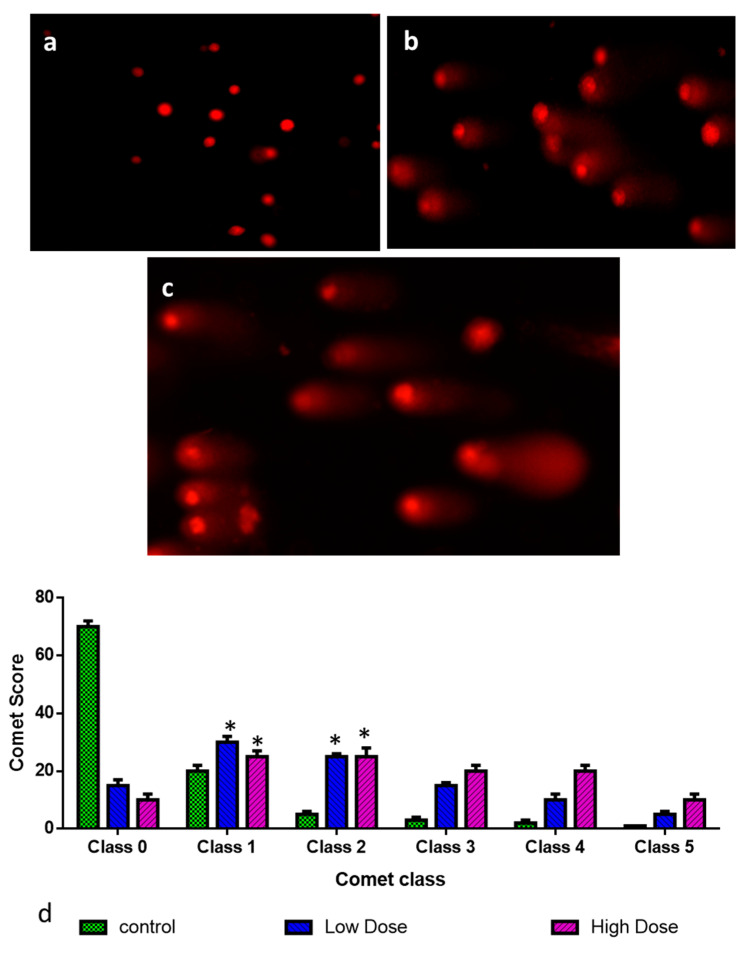
MDA-MB-468 cells were administered albumin visnagin NPs to measure DNA damage. The cells were left untreated (**a**), low dosage treated with albumin visnagin NPs (**b**), and high dosage treated with albumin visnagin NPs (**c**). Comet assays, mean comet scores and comet classes were used to assess DNA damage in the control and albumin visnagin NPs exposed groups. A mean score was calculated from three independent samples (**d**) based on the tail size and shape, ranging from 0 (undamaged) to 4 (extremely damaged). Two independent experiments were conducted to obtain the mean ± SD. (* signifies *p* < 0.05 when compared with control).

**Figure 8 molecules-28-03228-f008:**
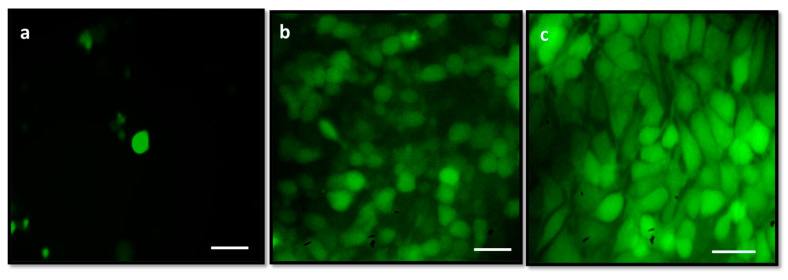
A fluorescence microscope image of endogenous ROS accumulation promoted by albumin visnagin NPs stained with DCFH-DA. Control (**a**), cells exposed to albumin visnagin NPs and IC25 (**b**) and IC50 (**c**). Image representing a triplicate experiment at 200X magnification (scale bar = 100 mm).

**Figure 9 molecules-28-03228-f009:**
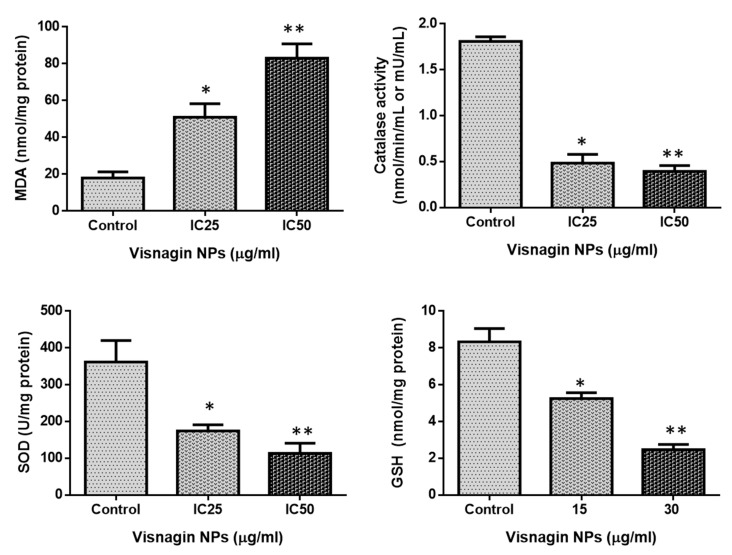
MDA-MB-468 cells treated with albumin or visnagin NPs for the treatment of CAT, SOD, GSH, and MDA. Each unit of CAT activity is equal to 1 mol H_2_O_2_ decomposed every second. It can be calculated that one unit of SOD activity equals the quantity of protein that inhibits 50% of the super oxygen radical oxidation of hydroxylamine to nitrite by superoxide radicals. A mean and standard deviation are presented for each of the values. It is significant to see the differences in bars with different letters, ** *p* < 0.01, * *p* < 0.05, between the treated and control groups.

**Figure 10 molecules-28-03228-f010:**
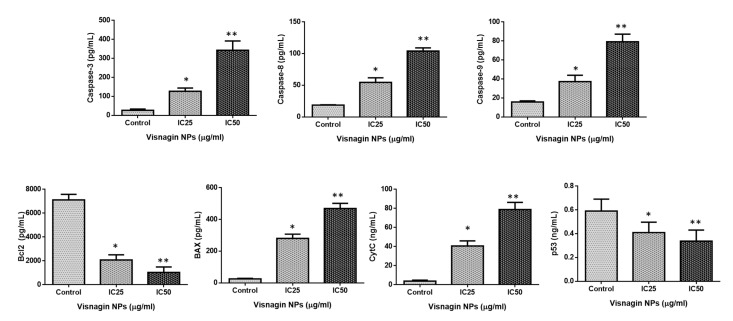
Albumin visnagin NPs inhibit cell proliferation in MDA-MB-468 cells (b) and promote apoptosis. Activities of caspase-3, 8, 9, Bax, Bcl-2, CytC, and P53 in both cells were determined using kits. All the experiments were performed in triplicates. The data are given as mean ± SD of triplicates. One-way ANOVA was utilized to analyze the values. ** *p* < 0.01 and * *p* < 0.05 compared with control.

**Figure 11 molecules-28-03228-f011:**
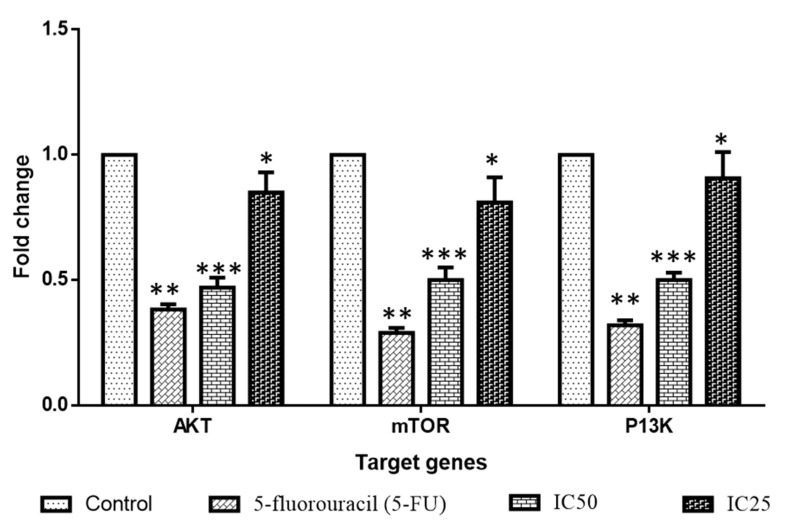
Effect of albumin and visnagin NPs on the PI3K/AKT/mTOR pathway in MDA-MB-468 cells. Values were presented as mean ± SD of the three samples. Data are analyzed by one-way ANOVA and Tukey postdoc assay using SPSS software. *** *p* < 0.0001 compared with control, ** *p* < 0.005 compared with control, * *p* < 0.05 compared with control.

**Table 1 molecules-28-03228-t001:** List of primers utilized in the study.

Genes	Primers	Sequence
PI3K	Forward Reverse	5′-AACACAGAAGACCAATACTC-3′ 5′-TTCGCCATCTACCACTAC-3′
AKT	Forward Reverse	5′-AGAAGCAGGAGGAGGAGGAG-3′ 5′-CCCAGCAGCTTCAGGTACTC-3′
mTOR	Forward Reverse	5′-AGGCCGCATTGTCTCTATCAA-3′ 5′-GCAGTAAATGCAGGTAGTCATCCA-3′

## Data Availability

Based on the request, all authors agreed to share their research data.

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
