# Peer review of "Therapeutic Potential of Albumin Nanoparticles Encapsulated Visnagin in MDA-MB-468 Triple-Negative Breast Cancer Cells"

_molecules, 2023, doi:10.3390/molecules28073228_

Round 1

Reviewer 1 Report

The paper “An in vitro study of albumin Visnagin nanoparticles on human 2 breast cancer cells (MDA-MB-468): Synthesis, characterization, antibacterial and anticancer properties” is interesting and a well-designed study about synthesis and some biological properties of nanoparticles of albumin and furanochromone visnagin. The manuscript needs some technical changes prior to its publication:

1.            The names of plant species and microorganisms should be written in italic.

2.            The authors must show the FTIR spectra of visnagin for comparison with the FTIR spectra of albumin visnagin nanoparticles. This is crucial for the conclusion of the nanoparticles contain incorporated visnagin in their structure.

3.            Figures 2 and  3 are not appropriately described in their caption. There are some mistakes. 

4.            Figure 4b does not have an explanation in the Figure 4 caption.

5.            The figure caption for Figure 5 is not appropriate. Explain specifically what is shown in pictures a and b.

6.            The figure caption for Figure 5 is not appropriate, it looks like a result description.

7.            Figure 8a does not have an explanation in the Figure 8 caption.

8.            In the Figure 9 caption there is a description for picture B but B is not marked in Figure 9.

9.            4.1. Chemicals: The names of suppliers for chemicals should be stated.

10.          All methodologies used in this study do not have cited references.

11.          The methodology of the determination of antibacterial and antifungal activities of Albumin Visnagin NPs is not appropriately described.

Author Response

Answers to Reviewer’s Comments

The authors thank the reviewer and the editor for the valuable comments and suggestions of the manuscript, which have greatly improved the quality of the Manuscript. The following is the comment and the revisions made:

Reviwer-1

  1. The names of plant species and microorganisms should be written in italic.

Answer: Authors thanks to the reviewer’s for valuable suggestion. As per reviewer’s suggestion we changed plant species and microorganisms in italic.

  1. The authors must show the FTIR spectra of visnagin for comparison with the FTIR spectra of albumin visnagin nanoparticles. This is crucial for the conclusion of the nanoparticles contain incorporated visnagin in their structure.

Answer: In the literature, many researchers have done FTIR spectra for Visnagin. As a result, we do not use the FTIR spectrum. We were able to match our albumin visnagin nanoparticle spectrum results with early literature. 

  1. Figures 2 and 3 are not appropriately described in their caption. There are some mistakes.

Answer: We changed the figure captions for Figures 2 and 3 (TEM and FESEM) in the revised manuscript. 

  1. Figure 4b does not have an explanation in the Figure 4 caption.

Answer: In the revised manuscript, we clearly explain the Fig. 4 caption.

  1. The figure caption for Figure 5 is not appropriate. Explain specifically what is shown in pictures a and b.

Answer: As per another reviewer’s suggestion we removed the antimicrobial study from this revised manuscript.

  1. The figure caption for Figure 5 is not appropriate, it looks like a result description.

Answer: As per another reviewer’s suggestion we removed the antimicrobial study from this revised manuscript.

  1. Figure 8a does not have an explanation in the Figure 8 caption.

Answer: As per reviewer’s suggestion we included Figure 8a in the Figure 8 caption.

  1. In the Figure 9 caption there is a description for picture B but B is not marked in Figure 9.

Answer: As per reviewer’s suggestion we included Figure 9b in the Figure 9.

  1. 4.1. Chemicals: The names of suppliers for chemicals should be stated.

Answer: The names of suppliers for chemicals were stated in the revised manuscript and the changes were marked in red text.

  1. All methodologies used in this study do not have cited references.

 Answer: All methodologies used in this study were cited with references and the changes were marked in red text.

  1. The methodology of the determination of antibacterial and antifungal activities of Albumin Visnagin NPs is not appropriately described.

Answer: As per another reviewer’s suggestion we removed the antimicrobial study from this revised manuscript.

Reviewer 2 Report

1-     The introduction and discussion should be focused more on the observations and novelty of this study and supported with related references. The authors may use the following references

a. https://doi.org/10.1016/j.jmrt.2020.10.021

b. https://doi.org/10.1016/j.jmrt.2020.10.021c. https://doi.org/10.3390/pharmaceutics130710982. Increase the resolution and quality of figures (SEM and XRD) . Scale bars are required for Fig. 7 and 113. Ic25 and Ic 50 not clear . This point needs to be clarified. Are author used the Ic50 for 24 or 48 0r 72 H?4. Classes for comet assay are required 5. The size particle distribution histogram and statically analysis is required.  6. Many typos and mistakes were observed. Scientific names for bacteria must be in italics. Thus, English language editing is required.7. Reagents with materials must be added with their origin (city, country). 8. More concluding remarks must be also added.

Author Response

Answers to Reviewer’s Comments

The authors thank the reviewer and the editor for their valuable comments and suggestions on the manuscript. These comments have greatly improved the quality of the manuscript. The following is the comment and revisions:

Reviwer-2

1-     The introduction and discussion should be focused more on the observations and novelty of this study and supported with related references. The authors may use the following references

  1. https://doi.org/10.1016/j.jmrt.2020.10.021
  2. https://doi.org/10.1016/j.jmrt.2020.10.021
  3. https://doi.org/10.3390/pharmaceutics130710982.

Answer: We included the reference in the revised manuscript and changes were marked in red text.

  1. Increase the resolution and quality of figures (SEM and XRD).

Answer: In the revised manuscript, we included in the high-resolution image.

  1. Scale bars are required for Fig. 7 and 11

Answer: We included the scale bar in the revised manuscript and changes were marked in red text.

  1. Ic25 and Ic 50 not clear. This point needs to be clarified. Are author used the Ic50 for 24 or 48 0r 72 H?.

Answer: As per the reviewer’s suggestion, we clarified the Ic25 and Ic50 concentrations used in this study, and the changes were marked in red text.

  1. Classes for comet assay are required

Answer: As per reviewer’s suggestion we included the classes of comet and the changes were marked in red text.

  1. The size particle distribution histogram and statically analysis is required.

Answer: We observed fewer particles in the scale of 100 nm resolution, so that is the reason given in the average size of the NPs and the statistical value that there were fewer particles present in the TEM image because we did not include it in the particle sized distribution histogram diagram. 

  1. Many typos and mistakes were observed. Scientific names for bacteria must be in italics. Thus, English language editing is required.

Answer: We edited the English language, corrected typos and mistakes, and added italics to plant and bacteria names, marking the changes in red.

  1. Reagents with materials must be added with their origin (city, country).

Answer: The names of suppliers for chemicals were stated in the revised manuscript and the changes were marked in red text.

  1. More concluding remarks must be also added.

Answer: Our revised manuscript includes the conclusion, which is highlighted in red.

Reviewer 3 Report

Alsrhani et al. submitted an original paper regarding the anticancer effect of Visnagin nanoparticles on breast cancer cells. In general, the study is well-designed. However, the quality of the results is insufficient in its present form. 

Major issues:

  1. The main goal of the paper is to show the anticancer properties of the Visnagin NP, thus the results describing its antimicrobial action are irrelevant and should be removed from this paper. 
  2. The quality of the pictures is unacceptable (Figures 2, 7, 10, 11). Presented results should be transposed into number data, and representative images moved into the Supplementary section.
  3. In all X axes Concentration (ug/ml) should be changed into Visnagin NPs (ug/ml).
  4. Changes in PI-3K, Akt, mTOR expression do not correspond directly with the activity of signaling pathways. To assess the activity of the above-mentioned kinases, their phosphorylation status should be analyzed (by Western blot, for example).
  5. The results part should be divided into subsections regarding particular effects, for example Visnagin NPs affect breast cancer cell viability; Apoptosis induction in response to Visnagin NPs action etc.
  6. There needs to be more information in the M&M section regarding the source of the cell line and cell culture.
  7. Figure 6 - lack of statistical analysis

Minor issue;

The paper does not follow the MDPI template in the References section and should be updated. Also Fig. (1-7) should be changes to Figure; font 9 in figure legends.

Author Response

Answers to Reviewer’s Comments

The authors thank the reviewer and the editor for their valuable comments and suggestions on the manuscript. These comments have greatly improved the quality of the manuscript. The following is the comment and revisions:

Reviwer-3

  1. The main goal of the paper is to show the anticancer properties of the Visnagin NP, thus the results describing its antimicrobial action are irrelevant and should be removed from this paper.

Answer: The author thanks the reviewers for their valuable suggestions. We removed the section on antimicrobial activity based on the reviewer's recommendation

  1. The quality of the pictures is unacceptable (Figures 2, 7, 10, 11). Presented results should be transposed into number data, and representative images moved into the Supplementary section.

Answer: Thank you for the good suggestions that results should be transposed into number data and images moved into the supplementary section, but based on the journal guidelines and scientific readers interest to see the changes of representative images of each staining experiments for easy interpretation as well as conclusion of the studies. Due to the readers interest we still keep all the images in the manuscript with good quality pictures.

  1. In all X axes Concentration (ug/ml) should be changed into Visnagin NPs (ug/ml).

Answer: The X axes in the figure have been changed to Visnagin NPs (ug/ml) on the reviewer's suggestion.

  1. Changes in PI-3K, Akt, mTOR expression do not correspond directly with the activity of signaling pathways. To assess the activity of the above-mentioned kinases, their phosphorylation status should be analyzed (by Western blot, for example).

Answer: We do agree with the reviewer’s comments on the PI-3K, Akt, mTOR expression. In our preliminary in vitro studies, we have done the gene expression of PI-3K, Akt, mTOR targets in the cancer cells and we have expressed the values in the figure. For our ongoing and future animal experiments, we will do protein expression studies using western blot to quantify the PI-3K, Akt, mTOR expression and their phosphorylation status. Based on the current findings, we have concluded that our nano-composite have significant effects on the PI-3K, Akt, mTOR expression.

  1. The results part should be divided into subsections regarding particular effects, for example Visnagin NPs affect breast cancer cell viability; Apoptosis induction in response to Visnagin NPs action etc.

Answer: As suggested by the reviewer, we divided the results into subsections and marked the changes in red.

  1. There needs to be more information in the M&M section regarding the source of the cell line and cell culture.

Answer: As per reviewer’s suggestion, we added more information regarding the source of the cell line and cell culture in the M&M section.

  1. Figure 6 - lack of statistical analysis

Answer: Thank you for the suggestions. We have done the statistical analysis of all the results and same has been incorporated in the revised version of manuscript.

Minor issue;

The paper does not follow the MDPI template in the References section and should be updated. Also Fig. (1-7) should be changes to Figure; font 9 in figure legends.

Answer: We updated the MDPI template and font in the figure legends according to the reviewer's suggestions.

Reviewer 4 Report

In the manuscript “An in vitro study of albumin Visnagin nanoparticles on human breast cancer cells (MDA-MB-468): Synthesis, characterization, antibacterial and anticancer properties” the authors synthesize and characterize Albumin Visnagin nanoparticles and show their antimicrobial and anticancer activity against a breast cancer cell line.

This is a well-written paper, on a topic of interest to the readers of the Molecules journal and can be considered for publication after a minor revision.

My comments on this paper are as follows:

1.     The cytotoxicity of albumin visnagin NPs is higher after 24 h compared to a 48 or 72 h treatment. Could the authors explain this effect?

2.     In fig 14 authors compare the effect of NPs with 5-fluorouracil. Is there a reason why they did not use Doxorubicin as in the other studies reported here?   

3.     The authors should correct the statement that “No fluorescence was 282 seen in the control cells (fig. 11a)…” since in the figure can be noticed a slight fluorescence.

4.     The comment on page 14 from a previous reviewer should be deleted. However, the authors are encouraged to address this comment.

5.     The experimental sections should be carefully corrected, there are many incorrect notations, such as the number of cells 105 instead of 105     

Author Response

Answers to Reviewer’s Comments

The authors thank the reviewer and the editor for their valuable comments and suggestions on the manuscript. These comments have greatly improved the quality of the manuscript. The following is the comment and revisions:

Reviewer - 4

  1. The cytotoxicity of albumin visnagin NPs is higher after 24 h compared to a 48 or 72 h treatment. Could the authors explain this effect?

Answer: The author compares the cytotoxicity of albumin visnagin NPs at 24, 48, and 72 h and the text are highlighted in red in the revised manuscript.

  1. In fig 14 authors compare the effect of NPs with 5-fluorouracil. Is there a reason why they did not use Doxorubicin as in the other studies reported here?

Answer: MDA-MB-468 is a Triple-negative cancer line. In this type of cancer doxorubicin and 5-FU are both employed as drug regimes. Both can be used as positive drugs and, in this study, we used 5-FU as a positive control.

  1. The authors should correct the statement that “No fluorescence was 282 seen in the control cells (fig. 11a) …” since in the figure can be noticed a slight fluorescence.

Answer: As per the reviewer’s suggestion, we changed the sentence and the changes were marked in red text.

  1. The comment on page 14 from a previous reviewer should be deleted. However, the authors are encouraged to address this comment.

Answer: As per the reviewer’s suggestion, we corrected the typo error and the changes were marked in red text.

  1. The experimental sections should be carefully corrected, there are many incorrect notations, such as the number of cells 105 instead of 105

Answer: As per the reviewer’s suggestion, we corrected the typo error and the changes were marked in red text.

Round 2

Reviewer 2 Report

Accept and can be published 

Author Response

The author thanks the reviewer for accepting the revised manuscript

Reviewer 3 Report

I appreciate the Authors' effort and manuscript improvement. Still, minor errors have to be corrected:

Figures 5, 9, 13 - please, add statistics to the bar charts

Fig. 1- -4 should be Figure 1, Figure 2...

Figures 6 and 10 - scale 100m? Please, provide the proper value.

Author Response

Answers to Reviewer’s Comments

The authors thank the reviewer and the editor for the valuable comments and suggestions of the manuscript, which have greatly improved the quality of the Manuscript. The following is the comment and the revisions made:

Figures 5, 9, 13 - please, add statistics to the bar charts

Answer: The author thanks the reviewer for their valuable suggestion. As per the reviewer’s suggestion, we added statistics to the bar charts of Figures 5, 9, and 13.

Fig. 1- -4 should be Figure 1, Figure 2...

Answer: As per the reviewer’s suggestion we arranged the figures.

Figures 6 and 10 - scale 100m? Please, provide the proper value.

Answer: As per the reviewer’s suggestion we provided the proper value for the scale bar.